# Effectiveness of Workplace Interventions to Improve Health and Well-Being of Health and Social Service Workers: A Narrative Review of Randomised Controlled Trials

**DOI:** 10.3390/healthcare11121792

**Published:** 2023-06-17

**Authors:** Rahman Shiri, Risto Nikunlaakso, Jaana Laitinen

**Affiliations:** Finnish Institute of Occupational Health, P.O. Box 18, 00032 Helsinki, Finland; risto.nikunlaakso@ttl.fi (R.N.); jaana.laitinen@ttl.fi (J.L.)

**Keywords:** absenteeism, burnout, general health, job demands, job control, job satisfaction, happiness, occupational injuries, well-being, work ability, work engagement

## Abstract

Health and social service workers face high levels of workload and job stressors, which can affect their health and well-being. Therefore, it is important to evaluate the effectiveness of workplace interventions that aim to improve their mental and physical health outcomes. This review summarizes the findings of randomized controlled trials (RCTs) that examined the impact of different types of workplace interventions on various health indicators among health and social service workers. The review searched the PubMed database from its inception to December 2022 and included RCTs that reported on the effectiveness of organizational-level interventions and qualitative studies that explored barriers and facilitators to participation in such interventions. A total of 108 RCTs were included in the review, covering job burnout (N = 56 RCTs), happiness or job satisfaction (N = 35), sickness absence (N = 18), psychosocial work stressors (N = 14), well-being (N = 13), work ability (N = 12), job performance or work engagement (N = 12), perceived general health (N = 9), and occupational injuries (N = 3). The review found that several workplace interventions were effective in improving work ability, well-being, perceived general health, work performance, and job satisfaction and in reducing psychosocial stressors, burnout, and sickness absence among healthcare workers. However, the effects were generally modest and short-lived. Some of the common barriers to participation in workplace interventions among healthcare workers were inadequate staff, high workload, time pressures, work constraints, lack of manager support, scheduling health programs outside work hours, and lack of motivation. This review suggests that workplace interventions have small short-term positive effects on health and well-being of healthcare workers. Workplace interventions should be implemented as routine programs with free work hours to encourage participation or integrate intervention activities into daily work routines.

## 1. Introduction

Health and social service workers are often exposed to high levels of stress and heavy workloads in their work, especially since the COVID-19 pandemic began [1,2,3]. Even though the pandemic is easing, they still face long working hours and delayed medical care [4]. In Western countries, the demand for health and social services is increasing due to ageing populations and workforce shortages [5], which can lead to more stress and workload for the workers. High levels of stress and workload can negatively affect the health and well-being of health and social service workers, so they are at increased risk of burnout [6,7,8], stress [3,8], insomnia [3], anxiety [3], depression [3], impaired work ability [9], and occupational injuries [10,11]. Between 25% and 50% of healthcare workers report burnout [6,7,12] and psychological distress [13,14], and 25% report a decline in their work ability [9]. Psychosocial work stressors can also affect work ability [15], and job burnout can increase the risk of musculoskeletal and mental disorders [16].

To tackle health and social service workers’ job stressors, it is important to implement effective interventions in the workplace that can reduce physical and psychosocial risk factors, improve work conditions, and enhance workers’ health and well-being. To date, multiple systematic reviews of the effectiveness of workplace interventions to improve the health and well-being of healthcare workers have been published. One type of intervention that has been widely used among health and social service workers is mindfulness-based cognitive therapy, particularly to reduce job Burnout Several systematic reviews have evaluated the effectiveness of mindfulness-based practices for health and social service workers, but the results are mixed. Some systematic reviews of clinical trials have found small positive effects of mindfulness-based interventions on reducing burnout [17,18,19], increasing well-being [20], and improving job performance [21] among healthcare workers, while other systematic reviews have reported inconsistent or null effects of mindfulness-based interventions on burnout [22,23], perceived general health [21], or physical health [20]. Moreover, the effects of other types of workplace interventions on the health and well-being of health and social service workers, particularly the effectiveness of workplace interventions in preventing occupational injuries and absenteeism, are still unclear. More research is needed to compare the effectiveness of different workplace interventions to improve the health and well-being of health and social service workers.

There have been challenges and barriers to participation in workplace interventions, so the participation rate has been low to moderate [24]. To date, the barriers and facilitators to participation in workplace interventions among health and social service workers have not been studied well. This narrative review of randomized controlled trials (RCTs) aimed to synthesize the evidence on workplace interventions that can improve the health and well-being of health and social service workers. Specifically, we addressed the following objectives: (1) to identify workplace programs that have been shown to enhance work ability, well-being, perceived general health, work performance, and job satisfaction among health and social service workers; (2) to identify workplace programs that have been shown to reduce psychosocial stressors, job burnout, sickness absence, and occupational injuries among health and social service workers; and (3) to identify the barriers and facilitators to participation in workplace interventions among health and social service workers.

## 2. Methods

Using a combination of MeSH (medical subject headings) terms and text words (Table 1), R.S. conducted a PubMed search of articles dating from its inception through December 2022. The searches were restricted to English-language studies and did not impose any limits on the age or sex of participants. Additionally, R.S. manually searched the reference lists of the reviews [17,18,19,21,23,25,26] on these topics for more relevant studies. R.S. screened the titles, abstracts, and full texts of the articles that met the inclusion criteria. The review included quantitative studies on the effectiveness of workplace interventions for health or social service workers in improving work ability, well-being, perceived general health, work performance, and job satisfaction or in reducing burnout, sickness absence and occupational injuries and accidents. The review also searched for qualitative studies that explored challenges, barriers, enablers, and facilitators to participation in workplace interventions for health or social service workers. The following search string was used and limited to the outcomes of interest: (challenges[tiab] OR barriers[tiab] OR enablers[tiab] OR facilitators[tiab]) AND (“qualitative research”[tiab] OR “qualitative study”[tiab]). The review only included individually or cluster randomized controlled trials as eligible quantitative studies. The review excluded randomized clinical trials without a control group, non-randomized controlled trials, and randomized controlled trials conducted among informal caregivers. The review qualitatively synthesized the results of the included quantitative studies. The review did not evaluate the methodological quality of the included studies.

## 3. Results

R.S. screened the titles and abstracts of 3271 reports from PubMed search and selected 220 studies for full-text review (Figure 1). We included 108 studies (114 reports, 2 reports from 6 studies) in the review and excluded 112 studies. We excluded 92 trials because they were not randomized controlled trials or they did not have a control group. Of the excluded studies, informal caregivers participated in 10 RCTs and medical, nursing, or pharmacy students or nursing academicians participated in seven RCTs. One trial only reported qualitative results, and two RCTs had only one session of 20–30 min intervention and measured the outcomes of interest right after a single session. The sample size of included RCTs varied from 21 to 2182 individuals. The 108 included RCTs were conducted in the USA (n = 26), Germany (n = 10), the Netherlands (n = 10), Sweden (n = 9), Denmark (n = 6), Turkey (n = 6), China (n = 5), Spain (n = 5), Australia (n = 4), Japan (n = 4), Canada (n = 3), Norway (n = 3), the UK (n = 3), Finland (n = 2), Malaysia (n = 2), Vietnam (n = 2), Belgium (n = 1), Chile (n = 1), France (n = 1), Greece (n = 1), Hong Kong (n = 1), Ireland (n = 1), South Korea (n = 1), and Switzerland (n = 1). Our additional search on factors that influenced participation in workplace interventions identified 760 publications. Out of these, we included five studies in the review.

### 3.1. Work Ability

We reviewed 12 RCTs (14 reports) that tested the impact of workplace interventions on work ability (Appendix A). Female healthcare workers who engaged in supervised high-intensity workplace strengthening exercises and group coaching reported better work ability than those who engaged in strengthening exercises at home [27]. However, the work ability index did not change during the follow-up for workers who engaged in strengthening exercises at work, and the small difference (Cohen’s d = 0.24) between the intervention and control groups was because of worsening work ability for workers who engaged in strengthening exercises at home [27]. Nurses with a musculoskeletal disorder who received work-related psychosocial coaching had better work ability regarding physical demands at 11- and 22-week follow-ups [28], but this effect disappeared at 24-month follow-ups [29]. An intervention to increase physical activity and lower fat and salt intake improved work ability index for nursing assistants [30], and 10-week customized aerobic fitness and strength exercises improved current work ability for healthcare workers with previous work-related back or upper body musculoskeletal problems [31].

Female nurses who participated in a 4-week workplace program of health education, yoga training, and individualized nutrition counselling reported higher work ability index after 3 months [32]. A combination of physical training, participatory ergonomics, and cognitive behavioural training slightly improved the current work ability of elder-care personnel [33]. Participatory ergonomic modifications slightly improved the work ability of young dental professionals only at 9- and 12-month follow-ups, not at 3- and 6-month follow-ups [34]. Football or Zumba training for an hour, 2–3 times per week for 40 weeks, did not affect the work ability of female hospital employees [35]. Other interventions that did not improve perceived work ability among healthcare workers were: a life-management strategy to cope with job demand and/or increase job resources [36], a small-group intervention based on theory of successful ageing for mental health promotion [37], an intervention with diet, strengthening and fitness exercises, and cognitive behavioural training [38], and integration of health protection and health promotion into continuous improvement system (Kaizen) [39].

### 3.2. Well-Being

We reviewed 13 RCTs (14 reports) that tested the impact of workplace interventions on well-being (Appendix A). The following interventions enhanced well-being among healthcare workers: supervised high-intensity workplace strengthening exercises plus group coaching [40]; 6-week supervised workplace exercises involving stretching, strengthening, aerobic and balance exercises [41]; an intervention to enhance coping with job demand and/or to boost job resources [36]; unguided digital mindfulness-based self-practices [42]; and an 8-week mindfulness-based stress reduction intervention [43]. However, the effect size was either small [42], or the intervention only improved well-being in employees with low job control at baseline and not in those with high job control [36].

The following interventions did not enhance well-being among healthcare workers: mindful colouring of mandalas for 10 days [44]; a 40-week football or Zumba training course for an hour per session, 2–3 times per week [45]; an intervention of 22 text messages about factors associated with burnout in a period of 10 months [46]; a small-group intervention based on theory of successful ageing for mental health promotion [37]; an 8-week relaxation meditation course [47]; in-person mindfulness resilience training or smartphone-delivered resilience training [48]; an 8- to 12-week online multi-component positive psychology intervention of psychoeducation and positive psychology exercises [49]; and a 3-h community resiliency training course (psychoeducation and sensory awareness skills) [50].

### 3.3. Perceived General Health

Nine RCTs assessed how workplace interventions affect perceived general health (Appendix A). A 40-week Zumba training program (one hour per session and 2–3 sessions per week) [45], an 8-week cultural participation program (films, concerts, art exhibitions, or choir singing) [51], and a 5-day course on role-playing/simulation and discussion of strategies to deal with work stressors and stressful situations [52] improved self-rated health among healthcare workers, but the effects were small [45,51]. A 12-week mobile wellness program to promote physical activity and improve sleep quality enhanced wellness among female nurses working rotating shifts [53]. A 10-week peer support intervention for work-related stress and burnout improved perceived general health among health and social service workers [54].

A multidisciplinary program that included exercise, nutrition, ergonomics, and psychological interventions [55], a 9-month integrated health program that consisted of exercise, stress management, health information and an examination of workplace to cope with the job [56], an open-rota intervention to improve work scheduling [57], or integration of health promotion and health protection into continuous improvement system (Kaizen) [39] had no beneficial effect on perceived general health among healthcare workers.

### 3.4. Job Burnout

We reviewed 56 RCTs that evaluated the impact of workplace interventions on burnout (see Appendix A). Some of these interventions showed positive results in reducing burnout and its components. Social service workers who received an 8-week brief stress management intervention reported lower levels of overall burnout, emotional exhaustion and depersonalization, and higher levels of personal accomplishment [58]. Similarly, ICU nurses with burnout who participated in an 8-week mindfulness-based intervention experienced reductions in emotional exhaustion and depersonalization and increases in personal accomplishment at 1- and 3-month follow-ups [59]. Other studies found that a 4-week brief mindfulness-based intervention decreased emotional exhaustion and depersonalization at three months follow-up among primary care physicians [60], a 6-week mindfulness stress reduction training reduced burnout at six weeks and three months post-intervention among nurses working at elderly care centres [61], and a 4-week brief mindfulness-based stress reduction intervention reduced burnout among healthcare workers [62]. A 6-week mindfulness-based yoga exercises also reduced burnout among healthcare workers [63].

However, the effects of mindfulness-based interventions on burnout were not consistent across all RCTs. A tailored mindfulness-based program had reduced burnout among resident physicians at a 6-month follow-up but not at 2- or 12-month follow-ups, and the effect size was small [64]. A 10-week mindfulness education and exercise program only marginally improved burnout among primary care medical practitioners [65]. An 8-week mindfulness-based stress reduction program that included psychoeducation and exercises improved personal accomplishment but had no effects on emotional exhaustion and depersonalization among resident physicians [66], while another RCT [67] showed the opposite effect: a beneficial effect on emotional exhaustion but not on depersonalization and personal accomplishment among healthcare professionals. Moreover, an 8-week mindfulness-based stress reduction intervention reduced overall burnout, emotional exhaustion, and depersonalisation but not personal accomplishment [68], while another trial showed beneficial effects on emotional exhaustion and depersonalisation but not on depersonalisation [69] among healthcare workers. Other RCTs have found no effects from unguided digital mindfulness-based self-practices [42], an 8-week mindfulness-based cognitive training [70], or in-person mindfulness resilience training or smartphone-delivered resilience training [48] on burnout or its components among healthcare workers. A brief mindfulness-based intervention (4 or 5 weeks) also failed to reduce emotional exhaustion or depersonalization [71,72] or improve personal accomplishment [71] among healthcare professionals. However, one of these RCTs used only two items of the Maslach Burnout Inventory to measure these outcomes [72]. Among resident physicians, one small trial showed that a tailored mindfulness-based cognitive training program improved emotional exhaustion and depersonalization [73]. However, this trial used a 2-item screening measure, while another small trial that used a 9-item Maslach Burnout Inventory-Human Service Survey showed no effect [73].

Several interventions have been shown to reduce burnout among healthcare workers. These include: improving working conditions [74], practicing progressive muscle relaxation exercise for three weeks [75], attending a 5-day course on role-playing/simulation, and coping strategies for work stressors and stressful situations at the workplace [52], and participating in a 4-week laughter yoga program with stretching, relaxation, deep breathing, and laughter exercises [76]. A 6-month team-based support group that discussed and solved job stressors reduced emotional exhaustion and depersonalization [77], while a 12-week coping skill training course reduced emotional exhaustion [78,79]. Transcendental meditation for 20 min twice a day reduced emotional exhaustion among distressed healthcare professionals [80], and a 6-month guided web-based resilience-enhancing learning course reduced emotional exhaustion among neonatal intensive care unit healthcare workers [81]. Six sessions of positive psychology coaching in three months reduced burnout among primary care physicians, and the effect lasted up to six months follow-up [82]. A web-based professional group coaching session twice a week reduced emotional exhaustion among female resident physicians at a six-month follow-up [83]. An 8-week supervised yoga class [84] and management classes to improve communication skills, efficacy, emotional control, working skills, and conflict management [85] improved post-intervention emotional exhaustion and depersonalization but had no effect on personal accomplishment. An attention-based training intervention for seven weeks improved emotional exhaustion but not depersonalization and personal accomplishment [86]. A psychosocial coaching program for nurses with a musculoskeletal disorder reduced emotional exhaustion at 22 weeks [28], but not at 24 months [29]. An intervention to enhance psychological flexibility reduced emotional exhaustion at 3 months and increased personal accomplishment at 3 and 12 months for healthcare professionals caring for dementia patients [87]. A 9-month facilitated small-group curriculum with mindfulness, reflection, shared experience, and learning reduced high depersonalization but not overall burnout or emotional exhaustion for practicing physicians [88]. A 6-month self-facilitated small-group discussion on well-being reduced emotional exhaustion and depersonalization and increased personal accomplishment when measured as binary outcomes, but not when measured as continuous outcomes [89]. A psychosocial intervention reduced depersonalization for qualified nurses but not for unqualified nurses in a low-security mental health unit and had no effects on emotional exhaustion and personal achievement [90]. An 8-week personalised yoga intervention and a fitness intervention increased personal accomplishment for junior physicians, but personalised yoga also reduced depersonalisation more than fitness [91].

The effectiveness of some interventions for reducing burnout among healthcare workers is unclear. A recovery program based on cognitive–behavioural therapy and motivational strategies for sleep and recovery reduced burnout at post-intervention but not at 6-month follow-up [92]. A 10-week peer-support intervention had no effect on burnout among health and social service workers [54]. Moreover, none of the following interventions reduced job burnout among healthcare workers: a psycho-oncology training on supportive communication and a crisis intervention to address psychological problems for cancer patients [93], 22 text messages about factors associated with burnout over 10 months [46], mindful colouring of mandalas for 10 days [44], an 8-week spiritually based passage meditation [94], a 9-month facilitated group discussion [95], a one-page discussion guide for self-facilitated discussion [96], a 7-week coping training on cognitive and problem-solving coping strategies [97], a 7-week social support intervention [97], a 12-month psychological intervention based on problem-based learning and personnel empowerment [98], a 12-week authentic connections groups program to enhance resilience [99], a 3-h community resiliency training on psychoeducation and sensory awareness skills [50], an educational program to manage challenging behaviour [100], a 13-week training program in dementia care to improve caregivers’ knowledge and competencies [101], a 13-week relaxation training such as muscle relaxation, breathing relaxation or guided imaginary journeys [101], a compassion fatigue resiliency program [102], management change on psychosocial factors involving education and problem-solving discussions [103], and a 6-week self-guided internet intervention on self-efficacy or perceived social support [104].

### 3.5. Job Performance and Work Engagement

We reviewed 12 RCTs that evaluated the impact of workplace interventions on job performance or work engagement (Appendix A). A 10-week peer-support intervention for work-related stress and burnout among health and social service workers increased their work participation [54]. Six months of positive psychotherapy delivered via WeChat for nurses with burnout symptoms enhanced their job performance [105]. Three months of positive psychology coaching with six sessions [82], a 9-month facilitated small-group curriculum with 19 sessions that included mindfulness, reflection, shared experience, and learning [88], and an evidence-based medicine training program that comprised course and case method learning sessions [106] for healthcare workers boosted their work engagement. The benefits lasted up to 6 months [82] or 12 months [88] after the intervention. An 8-week mindfulness-based intervention that consisted of meditation, light yoga, and music for surgical intensive care unit personnel improved their work engagement [107]. A 12-week coping skill training for physicians working in emergency medicine improved their work engagement immediately after the intervention but not at three or six months later [79]. The following interventions did not show significant effects on work engagement among healthcare workers: a psychosocial competence intervention [108]; an eight- to 12-week online multi-component positive psychology intervention that consisted of psychoeducation and positive psychology exercises [49]; a one-page discussion guide for self-facilitated discussion [96]; a six-week smartphone stress management program based on cognitive behavioural therapy [109]; and a 6-week self-guided internet intervention that focused on self-efficacy or perceived social support [104].

### 3.6. Happiness and Job Satisfaction

We reviewed 35 RCTs (37 reports) that evaluated the effectiveness of workplace interventions on happiness and job satisfaction (Appendix A). Supervised high-intensity strengthening exercises at the workplace improved job satisfaction among female healthcare workers compared with strengthening exercises at home [40]. A psychosocial competence training course [108], short motivational mobile phone messages for three weeks [110], six sessions of positive psychology coaching within three months [82], interventions to improve working conditions [74], a 12-week coping skill training [78], a 5-day course on role-playing/simulation and discussion of practices to cope with work stressors and stressful situations at the workplace [52], and an open-rota intervention to improve work scheduling [57] improved job satisfaction among healthcare workers. The beneficial effect lasted up to six months follow-up [82]. An 8-week cognitive behavioural skill-building program improved job satisfaction at six months follow-up [111] but had no effect at post-intervention or 3-month follow-up [112]. On the other hand, an 8-week mindfulness-based stress reduction intervention improved job satisfaction within two weeks post-intervention but had no beneficial effect four months after the completion of the intervention [113], and a 12-week coping skill training increased job satisfaction at post-intervention but had no effect at 3- or 6-month follow-up [79]. A 4-week mindfulness-based stress reduction course guided by a website improved job satisfaction within eight weeks post-intervention; however, the effect was small [114].

The following interventions did not improve job satisfaction among healthcare workers: an intervention to promote physical activity and reduce fat and salt intake [30]; leadership skills training [115]; a multidisciplinary program that included exercise, nutrition, ergonomics, and psychological interventions [55]; a 9-month curriculum of mindfulness, reflection, shared experience, and learning in small groups [88]; an 8-week meditation program based on spirituality [94]; a 4-week intervention of brief mindfulness-based stress reduction [62]; a 3-month web-based stress management intervention [116]; a meaning-centred intervention (a coping strategy that focuses on finding meaning in work) [117]; a stress management intervention that involved lectures on effective coping strategies [113]; an 8-week mindfulness-based stress reduction intervention [118]; an 8- to 12-week online positive psychology intervention that consisted of psychoeducation and positive psychology exercises [49]; a person-centred intervention of dementia-care mapping to enhance the quality and effectiveness of care [119]; an educational program to manage challenging behaviour [100]; an enhanced primary healthcare intervention that integrated multidisciplinary care, continuous improvement of care delivery and organisational practices [120]; a 12-week psychosocial intervention that included resilience training, cognitive behavioural and solution-focused counselling [121]; a multidisciplinary care program for depression that involved procedures for screening, identification, diagnosis, treatment, and monitoring [122]; a leadership and management program that included an educational and support program for middle managers [123]; a 120-h ‘ePsychNurse.Net’ eLearning course [124,125]; a 6-month phone-based text message intervention to enhance medical knowledge [126]; a 6-month self-facilitated small group discussion on well-being [89]; and an evidence-based medicine training program comprising a course and case method learning sessions [106]. Moreover, a multidisciplinary end-of-life educational intervention that included two educational workshops and an educational booklet did not improve job satisfaction among health and social service workers [127]. A 24-week stress management and resilience training [128], and a 6-month guided web-based resilience enhancement course [81] did not boost happiness among healthcare workers.

### 3.7. Psychosocial Risk Factors

We assessed the impact of workplace interventions on psychosocial risk factors in 14 RCTs (see Appendix A). ICU nurses who participated in a 5-day course that involved role-playing/simulation and discussing strategies to deal with work stressors and stressful situations reported lower psychological demands, job strain (high psychological demands and low decision latitude), isostrain (job strain and low social support) and higher social support and decision latitude after six months [52]. These effects on job strain and isostrain lasted up to 12 months [52]. A working peer-support intervention for stress and burnout that lased 10 weeks reduced quantitative demands for health and social service workers [54], and a web-based stress management intervention for three months reduced workload for nurses [116]. A tailored mindfulness-based program had a small short-term effect on perceived job strain among resident physicians and reduced job strain after two months but not after six or 12 months [64]. A multi-faceted intervention that included integrated physical training, participatory ergonomics, and cognitive behavioural training reduced occupational lifting but did not affect physical exertion among elderly care personnel [33].

Several interventions aimed at improving the well-being of healthcare workers did not show significant effects on their work-related stress factors. A web-based career identity training for shift-working nurses did not change their workload, job control or reward from work [129]. Similarly, an intervention to enhance coping with job demands and/or to increase job resources [36], or a small-group intervention based on theory of successful ageing for the promotion of mental health [37] did not increase job control among nurses. An intervention to increase physical activity and reduce fat and salt intake had no effects on effort, reward, or effort/reward ratio [30], and a 9-month integrated health program that included exercise, stress management, health information and workplace examination to cope with the job had no effects on demands, control, or effort reward imbalance [56] among healthcare workers. Moreover, an 8-week brief stress management intervention to increase psychological flexibility among social workers did not influence their perceived psychological demands and control [58]. Additionally, an educational program to deal with challenging behaviour [100], a multidisciplinary care program for depression that consisted of procedures for screening, identification, diagnosis, treatment, and monitoring [122], or a management change on psychosocial factors that included an initial education about stressors and stress management and followed by practical problem-solving discussions [103] did not decrease job demands [100,103,122] or increase job control [103] among healthcare workers.

### 3.8. Sickness Absence

We reviewed 18 RCTs that evaluated how different interventions in the workplace can reduce the number of days that healthcare workers are absent due to illness (see Appendix A). We found that some interventions were effective, while others were not. Healthcare workers who had frequent non-specific lower back pain benefited from a combination of neuromuscular exercises and counselling for six months, but not from either one alone [130]. Nurses and nursing aides who engaged in strengthening, endurance, and coordination exercises on average six times per week during work hours had fewer days off work because of back pain [131], and nurses and nursing aides who received 2 weeks of acceptance and commitment therapy had fewer days off work because of work-related injuries [132]. Home care workers with non-specific lower back pain who wore a lumbar support belt on working days had fewer self-reported days off work because of lower back pain, but this did not affect the number of registered all-cause sickness absence days [133].

We also found that the following interventions did not reduce the number of sickness absence days among healthcare workers: Online mindfulness programs without guidance [42]; a personalized mindfulness-based program [64]; a multidisciplinary program that included exercise, nutrition, ergonomics and psychology [55], text messages about burnout prevention [46]; tailored aerobic fitness and strength exercises for 10 weeks [31]; a multi-faceted intervention that included physical training, participatory ergonomics, and cognitive behavioural training [33]; a program that included diet, strengthening and fitness exercises, and cognitive behavioural training [38]; a 6-month light aerobic, strengthening and stretching exercises for an hour per week [134], a 5-week physical and behavioural preventive intervention [135]; a multifaceted intervention that included participatory ergonomics, healthy lifestyle promotion and a tailored case management [136]; an 8-week educational and individualized program to manage sleep and the consequences of shift work [137]; a 9-month integrated health program that consisted of exercise, stress management, health information and an examination of workplace to cope with the job [56]; education of patient transfer technique [138]; patient transfer technique education combined with supervised aerobic fitness and strength training [138]; and integration of health promotion and health protection into continuous improvement system (Kaizen) [39].

### 3.9. Occupational Injuries

We evaluated the impact of workplace interventions on occupational injuries using three RCTs (four reports) (Appendix A). Needle safety devices plus a workshop reduced self-reported needle stick injuries among healthcare workers, but not officially registered injuries, compared with workshop only or no intervention [125,139]. A leadership and management program with an educational and support program for middle managers did not affect fall with injury among aged care home workers [123]. Likewise, 2 weeks of acceptance and commitment therapy did not change the work-related injury rate among nurses and nursing aides [132].

### 3.10. Challenges and Barriers

Nursing assistants reported that insufficient breaks, patientcare tasks interrupting breaks, working beyond their scheduled shift, and increased work responsibility prevented them from participating in physical activity interventions [30]. Nurses also faced barriers such as lack of adequate equipment, staff shortages, low motivation in older nurses, high workload, time constraints, and insufficient training to use workplace equipment for preventing musculoskeletal disorders [140]. Female home care helpers and assistants mentioned rigid care work, unsupportive team leaders, trainings not organized during work hours, high physical and mental workload, and intervention intensity hindered their participation in a workplace health promotion program [141]. They said that taking part in an intervention meant working overtime or having their colleagues perform their work [141]. Time, knowledge and interest shortage, and negative experience were reported as barriers to participating in mindfulness-based interventions [142]. Moreover, ICU nurses and nursing assistants did not join a psychological intervention because of work limitations or unwillingness to participate during their free time [98].

## 4. Discussion

This review summarizes the findings of 108 RCTs that evaluated the impact of workplace interventions on employee health outcomes. The majority of the RCTs focused on job burnout and job satisfaction, while occupational injuries were rarely studied. The RCTs also mainly targeted healthcare workers, with only three trials involving social workers. According to our qualitative synthesis, workplace exercises such as a combination of strengthening exercises and aerobic fitness prevented the deterioration of work ability and workplace exercises and mindfulness-based practices improved well-being. A combination of neuromuscular exercise and counselling, or a combination of strengthening exercises and endurance/coordination exercises, reduced the rate of sickness absence due to musculoskeletal disorders. Multiple workplace interventions had small but positive effects on work performance, job satisfaction, psychosocial risk factors, and job Burnout Only three RCTs evaluated the impact of workplace interventions on occupational injuries and none of the interventions was effective. To implement a successful workplace health program, employers can either offer a health intervention as a regular program with free work hours to encourage employee participation, or integrate intervention activities into daily work routines, such as regular work meetings.

Job burnout is a prevalent and serious health issue among healthcare workers, affecting between 26% and 55% of them, according to different studies [6,7,12]. Burnout in healthcare workers compromises quality of care and patient safety [143]. More than half of the RCTs in this review assessed the effectiveness of workplace interventions in lowering employee job Burnout The results of this review suggest that mindfulness-based interventions are helpful in reducing the level of burnout among healthcare professionals; however, the effects are modest. A systematic review of clinical trials reported mixed results found by RCTs on the impact of mindfulness-based interventions on burnout among healthcare workers [22]. Systematic reviews of clinical trials among nurses [17,18] and a systematic review and meta-analysis of interventional studies among physicians [19] found small positive effects of mindfulness-based interventions on Burnout However, in another systematic review, only one of four clinical trials showed improved burnout symptoms among healthcare workers after brief mindfulness-based interventions [23], and in another systematic review, only 11 out of 25 interventional and observational studies showed beneficial effects of mindfulness-based interventions on burnout among healthcare workers [21]. A meta-analysis of four controlled clinical trials showed beneficial effects of meditative interventions on emotional exhaustion and personal accomplishment among healthcare professionals [144]. Moreover, a meta-analysis of nine controlled clinical trials found a small effect of mindfulness-based interventions on post-intervention burnout among healthcare professionals and trainees, but a meta-analysis of two trials that measured burnout at follow-up did not show a relationship between mindfulness training and burnout [20]. Additionally, a meta-analysis of seven controlled clinical trials found that coping strategies reduce burnout among nurses [145].

The current review suggests that mindfulness-based interventions and workplace exercises improve employee well-being. According to a meta-analysis of 24 controlled clinical trials, mindfulness-based interventions had a small positive impact on well-being outcomes immediately after the intervention, and a meta-analysis of nine trials found a similar effect at follow-up for healthcare professionals and trainees [20]. Additionally, a meta-analysis of five RCTs found that resilience training had a moderate positive impact on well-being immediately after the intervention and a meta-analysis of three RCTs found a small positive impact at short-term (≤3 months) follow-up for nurses [146]. Mindfulness interventions may enhance mindfulness skills to focus on the present moment and reduce stress-related rumination, which may improve the ability to detach from work, a key psychological recovery experience [147]. A systematic review of 13 interventional and observational studies reported that six studies found a positive effect from mindfulness-based interventions on job performance for healthcare workers [21]. However, a meta-analysis of three controlled clinical trials found no effect of mindfulness-based interventions on physical health for healthcare professionals and trainees [20]. Moreover, only three out of seven interventional and observational studies included in a systematic review reported a positive effect of mindfulness-based interventions on general health for healthcare workers [21]. Longer and more intensive interventions tend to be more effective than shorter and less intensive ones [147].

Needlestick and sharps injuries are common among nurses, and working long hours, overtime and rotating shifts can increase the risk of these injuries [148]. However, only three of the 108 RCTs included in this review evaluated the effectiveness of workplace interventions in preventing occupational injuries and none of them found any significant effect.

Workplace health promotion programs can enhance the physical and mental health of workers, but they often have low participation rate [24]. No significant differences in participation rates were found by age, education, or income level, but women were more likely to participate in educational and multi-component programs than men [24]. The current review identified lack of staff, heavy workload, time constraints, work-related obstacles, insufficient manager support, scheduling health programs outside work hours, and low motivation as factors that hinder participation in workplace health interventions. To encourage participation, employers can offer incentives, such as free work time or rewards, and address the other barriers by engaging employees in the planning and implementation of the programs.

The RCTs included in this review had some methodological limitations. Many of them used a cluster-RCT design, in which groups of individuals (e.g., wards, units, departments, clinics, hospitals, or worksites) are randomized to receive interventions [149]. This design avoids contamination and is appropriate when interventions target systems, programs, or worksites. However, many of the cluster-RCTs did not account for the clustering effect in their analysis. The clustering effect means that individuals within a group are more likely to have similar outcomes than individuals from different groups. This reduces the statistical power and precision of the trial and can lead to biased estimates [149,150]. Most of the RCTs also had a small sample size of healthcare workers, which limits the generalizability of their findings. The RCTs on sickness absence in this review had several limitations. They lacked enough statistical power, they only measured immediate effects of the interventions, they relied on self-reported data on the number of persons being on sick leave or number of sickness absence days in the past 3 to 12 months, and they did not perform full intention-to-treat analysis due to missing data from dropouts.

This review also had some limitations. A systematic review requires conducting literature searches in at least one large European database (such as Scopus, Embase) and one large American database (such as PubMed, Web of Science); however, the current review was a narrative review that only searched PubMed, which may have missed some relevant publications from other databases. The goal of this review was to find effective workplace programs for health and social service workers’ health. We think it is less likely that we overlooked many positive studies for workplace interventions. Most of the RCTs on workplace programs were conducted after 2000 and are indexed in both PubMed and Embase. However, some older RCTs may only be found in the Cochrane Library.

## 5. Conclusions

Workplace interventions can enhance work ability, well-being, perceived general health, work performance, and job satisfaction and can lower psychosocial risk factors, job burnout, and sickness absence among healthcare workers. However, these effects are modest and short-lived. They tend to fade away soon after the interventions are discontinued.

## Figures and Tables

**Figure 1 healthcare-11-01792-f001:**
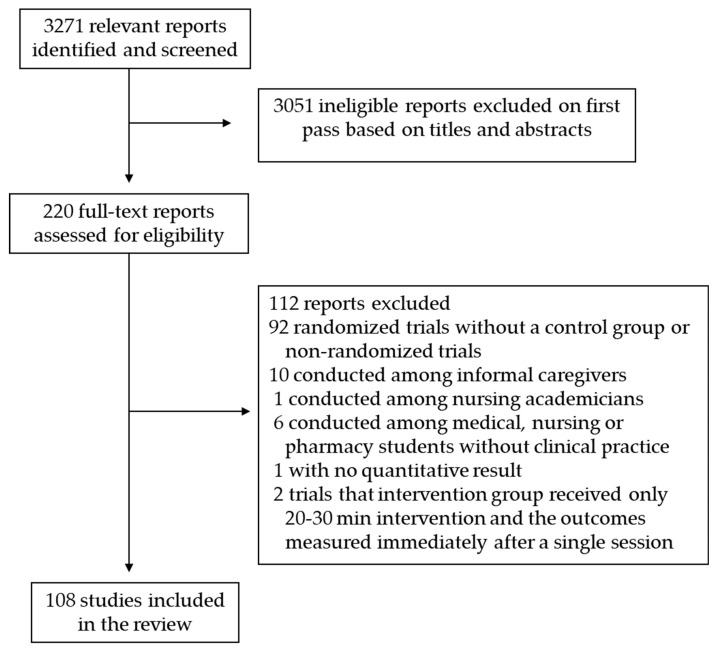
Flow diagram of the studies selection.

**Table 1 healthcare-11-01792-t001:** PubMed search conducted on 1 December 2022.

Search	Query	No of Items Found
#1	“healthcare sector”[Mesh] OR “allied health personnel”[Mesh] OR “health personnel”[Mesh] OR “home care agencies”[Mesh] OR “home care services”[Mesh] OR “social workers”[Mesh] OR “care sector workers”[tiab] OR social workers[tiab] OR “health worker*”[tiab] OR “healthcare worker*”[tiab] OR “healthcare worker*”[tiab] OR “health employee*”[tiab] OR “healthcare employee*”[tiab] OR “healthcare employee*”[tiab] OR “healthcare personnel”[tiab] OR “healthcare personnel”[tiab] OR “health professional*”[tiab] OR “healthcare professional*”[tiab] OR “healthcare professional*”[tiab] OR “medical care personnel”[tiab] OR “medical personnel”[tiab] OR “medical staff”[tiab] OR “medical professional*”[tiab] OR nurse[tiab] OR nurses[tiab] OR “social service staff”[tiab] OR “social services profession*”[tiab] OR “social service profession*”[tiab] OR “social care staff”[tiab] OR “social care profession*”[tiab] OR “social care provider*”[tiab] OR “medical staff, hospital”[Mesh] OR “community health workers”[Mesh] OR “community worker*”[tiab] OR welfare worker*[tiab] OR caseworker*[tiab] OR case-worker*[tiab] OR “public servant*”[tiab] OR almoner[tiab] OR “counselors”[Mesh] OR counsellor*[tiab] OR counsellor*[tiab] OR “mentors”[Mesh] OR “mentor*”[tiab] OR “school teachers”[Mesh] OR teacher*[tiab] OR “physical therapists”[Mesh] OR “physical therapist assistants”[Mesh] OR “occupational therapists”[Mesh] OR therapist*[tiab] OR physiotherapist*[tiab]	1,081,947
#2	workability[tiab] OR “work ability”[tiab] OR “work disability”[tiab] OR well-being[tiab] OR “well-being”[tiab] OR wellness[tiab] OR absenteeism[Mesh] OR “sick leave”[Mesh] OR “sick leave”[tiab] OR “sickness absence”[tiab] OR “sickness absenteeism”[tiab] OR “accidental injuries”[Mesh] OR “accidents, occupational”[Mesh] OR “occupational injuries”[Mesh] OR “workplace accident*”[tiab] OR “workplace injur*”[tiab] OR “occupational accident*”[tiab] OR “occupational injur*”[tiab] OR “self-perceived health”[tiab] OR “perceived health”[tiab] OR “self-rated health”[tiab] OR “job control”[tiab] OR “work control”[tiab] OR “work demand*”[tiab] OR “job demand*”[tiab] OR “job strain”[tiab] OR “work strain”[tiab] OR (effort[tiab] AND reward[tiab]) OR “work engagement”[Mesh] OR “work engagement”[tiab] OR “job satisfaction”[Mesh] OR “job satisfaction”[tiab] OR (satisfied[tiab] AND job[tiab]) OR “burnout, psychological”[Mesh] OR “burnout, professional”[Mesh] OR burnout[tiab] OR happiness[Mesh] OR happiness[tiab] OR “work performance”[Mesh] OR “work performance”[tiab] OR “job performance”[tiab]	261,777
#3	Clinical trial[pt] OR “clinical trials as topic”[Mesh] OR “clinical trial”[tiab] OR controlled clinical trial[pt] OR “controlled clinical trials as topic”[Mesh] OR “randomized controlled”[tiab] OR “randomised controlled”[tiab] OR “randomized trial”[tiab] OR “randomised trial”[tiab] OR “controlled trial”[tiab] OR “randomized controlled trials as topic”[Mesh] OR “non-randomized controlled trials as topic”[Mesh] OR “random allocation”[Mesh] OR “random allocation”[tiab] OR “controlled before-after studies”[mesh] OR “controlled before-after study” [tiab] OR “cross-over studies”[mesh] OR “cross-over study”[tiab] OR “crossover study“[tiab] OR “pseudo-experiment”[tiab] OR “pseudo-trial”[tiab] OR “quasi-experimental”[tiab] OR “quasi-experiment”[tiab]	1,552,610
#4	#1 AND #2 AND #3	3697
Final	#4 Filters: Humans, English	3271

MeSH, Medical Subject Headings, tiab, Title/Abstract, *, truncation symbol.

## Data Availability

No new data were created or analyzed in this study. Data sharing is not applicable to this article.

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
