# Peer review of "Effectiveness of Workplace Interventions to Improve Health and Well-Being of Health and Social Service Workers: A Narrative Review of Randomised Controlled Trials"

_healthcare, 2023, doi:10.3390/healthcare11121792_

Round 1

Reviewer 1 Report

The paper is interesting, well written well organized.

Authors in the discusson paragraph write that "No effective intervention was identified to reduce the risk of occupational injuries" but in relaity the studies taht dealt with this issue were only 4. This means rather that the evidence we have are not enough developed. The findings of this paper are interseting and maybe deserve justa  bit more elaboration/comments. For instance, it would be interesting to make comments about the specific settings were speciifc results were achieved; for instance, do you think that in a specifi setting were data were collected some factors of the work environment might have had an impact on the final results?

Author Response

Response. Thank you for your comments!

We have made the necessary revisions to the manuscript based on your feedback.

We cannot draw a definitive conclusion about the influence of the setting on the intervention outcomes. There is a lack of consistency among the results of different randomized controlled trials that implemented the same intervention in the same setting.

Author Response

Response. Thank you for your comments!
We have defined the terms "Mesh" and "tiab" in a footnote for clarity.
We agree that Table A1 is quite comprehensive. However, we think it is important to include it in the appendix, as it provides detailed information on the interventions that were evaluated. We faced difficulties in reducing the table without compromising its usefulness. Moreover, some readers may appreciate the thoroughness of the interventions.

We followed a rigorous screening process for the full text reports. Various interventions were implemented to improve job satisfaction and happiness among health and social workers. Some of these interventions may not seem relevant, but they were tested for their effects on employee happiness and job satisfaction. As expected, some interventions did not show positive outcomes.

We did not examine the influence of organizational governance and policy on the effectiveness of workplace interventions to improve health and well-being of health and social workers. Therefore, the findings of this review cannot support such a conclusion.

Reviewer 3 Report

The research is very interesting and presents an adequate structure and methodology. Similarly, the conclusions are well established. It is recommended to restructure the appendix to facilitate the understanding of the information and reduce its length.

Author Response

Response. We appreciate your positive feedback! We acknowledge that Table A1 is quite extensive. Therefore, we placed the table in the appendix. Reducing the table would be challenging. Furthermore, some readers might find the details of the interventions useful.

Reviewer 4 Report

The manuscript entitled 'Effectiveness of Workplace Interventions to Improve Health and Well-Being of Health and Social Service Workers: A Narrative Review of Randomised Controlled Trials' presents an extensive narrative review of RCTs focused on health and social service workers.
In general, the work is very rich in information, but it is rather fragmented. The impression of this reviewer is that the choice to include interventions on very different aspects (please, see point #2 of the search-query) is not particularly justified by the aims.
Also, it is not completelly clear to this reviewer how the Authors carried out the additional search (i.e., 'Furthermore, we performed an additional search to identify qualitative studies on challenges, barriers, enablers, and facilitators to participation in workplace interventions conducted among health or social service workers', page 2, lines 79-81). The Authors probably conducted this further investigation on the studies already included with the main search-query.
It is not very clear what this study adds to the numerous reviews already conducted (and cited in the Discussion). Probably, the Authors could make this explicit (e.g., at the beginning or end of the Discussion).
In this reviewer's opinion, this study would benefit from a paragraph summarising the types of interventions that were considered in the RCTs. The Authors partly do this in the Discussion, but the impression is that it would be useful to present them already in the Results (i.e., to facilitate the reading of thematic results).

Author Response

Response. Thank you for your comments!

The purpose of this review was to evaluate various workplace interventions on different dimensions of employee health and wellbeing. Furthermore, the review aimed to synthesize the effects of the same intervention on multiple employee health and wellbeing outcomes.

We applied an additional search string to retrieve qualitative studies on barriers and facilitators to participation in workplace interventions. We have added the following search string to the methods section on page 4.

“The review also searched for qualitative studies that examined challenges, barriers, enablers, and facilitators to participation in workplace interventions for health or social service workers. The following search string was used and restricted to the outcomes of interest: “(challenges[tiab] OR barriers[tiab] OR enablers[tiab] OR facilitators[tiab]) AND (“qualitative research” [tiab] OR “qualitative study” [tiab]).”

We have also added the following results on page 5.

“Our additional search on factors that influenced participation in workplace interventions yielded 760 publications. Out of these, we included five studies in the review.”

Summarizing the types of interventions would require multiple pages. It is not feasible to summarize them in a paragraph. Each trial implemented a different intervention in terms of type, frequency, dose, and duration. Interventions are described in detail in appendix Table A.

Round 2

Reviewer 4 Report

I would like to thank the Authors for the changes made to the manuscript. I basically agree with their answers.